# Variation analysis of growth traits and medicinal components in different provenances of *Polygonatum cyrtonema* based on heterogeneous garden experiment

Shiming Cheng[1], Ying Hu[2], Yan Cheng[3], Zhiyang Qian[2], Xiurong Xu[1], Xiashuo Lei[1], Xiaodeng Shi[1]*

1 Zhejiang Academy of Forestry, Hangzhou, Zhejiang, China, 2 Zhejiang A&F University, Hangzhou, Zhejiang, China, 3 Ningbo University of Finance and Economics, Ningbo, Zhejiang, China

* sxdpaper@163.com

## Abstract

*Polygonatum cyrtonema* is a valuable medicinal and edible plant whose sustainable utilization is challenged by wild resource depletion and germplasm degradation. This study established a multi-site provenance trial across three heterogeneous garden environments in Zhejiang Province, China, to evaluate 11 geographically diverse provenances. We systematically measured growth traits (plant height, stem diameter, leaf morphology) and medicinal components (polysaccharides, saponins, flavonoids, total phenolics), and applied combined ANOVA, correlation analysis, principal component analysis, and cluster analysis to quantify provenance variation patterns and environmental drivers. Results revealed highly significant differences (P<0.01) among provenances, sites, and their interactions for all traits. Substantial phenotypic (PCV: 7.41%–46.89%) and genotypic (GCV: 5.99%–44.92%) coefficients of variation were observed, with particularly high variation in polysaccharides and key growth traits, coupled with substantial provenance repeatability (0.63–0.99), indicated strong potential for selective breeding. Correlation analysis showed significant positive associations between growth traits and key medicinal components. Geo-climatic analysis identified distinct environmental drivers: saponin content increased with altitude and temperature, while flavonoid accumulation was promoted in drier conditions. Based on principal component analysis (cumulative contribution: 85.20%), Songyang (3.34) and Yunhe (2.98) provenances achieved the highest comprehensive evaluation scores. Cluster analysis further classified the provenances into three groups, with Songyang and Yunhe forming a distinct cluster characterized by superior growth and medicinal compound accumulation. These provenances are recommended as elite materials for breeding programs. This study provides a scientific basis for the selective breeding of *P. cyrtonema* and holds significant practical implications for

**Data availability statement:** All relevant data are within the manuscript and its Supporting information files.

**Funding:** This study was supported by the [Pioneer and Leading Goose +X" R&D Program of Zhejiang(2024C04007)]. Prof. Shiming Cheng, as the recipient of this funding project and the first author of this manuscript, was responsible for conceptualization, methodology, supervision, project administration.

**Competing interests:** The authors have declared that no competing interests exist.

enhancing the quality and efficiency of the understory economy and promoting the sustainable use of medicinal plant resources.

---

## 1. Introduction

*Polygonatum cyrtonema*, a perennial herb of the genus *Polygonatum* (Liliaceae), is natively distributed across the subtropical to temperate transitional zones of China, with its genuine producing areas concentrated in Anhui, Guizhou, and Zhejiang provinces, where it typically thrives in shaded understory habitats at elevations of 500–2100 m [1]. Its dried rhizome, designated as "*Polygonati Rhizoma*" in the Chinese Pharmacopoeia (ChP), is one of the three primary botanical sources (together with *Polygonatum sibiricum* and *Polygonatum kingianum*) for this official medicinal material [2]. Enriched with bioactive compounds including polysaccharides, flavonoids, and saponins, it demonstrates pharmacological effects such as antioxidant activity, anti-fatigue properties, anti-aging and dermatological benefits, hypoglycemic and hypolipidemic actions, as well as immune-enhancing capabilities [3,4]. Furthermore, its rhizomes have historically been used as a famine food due to their high carbohydrate content, which is comparable to staple crops like rice, earning it the revered name "Immortal's Surplus Grain" [5]. As a prominent medicinal and edible plant, *P. cyrtonema* holds strategic value in both the traditional Chinese medicine industry and the forest wellness economy. It is therefore considered a key understory economic crop in southern China, with significant industrial and economic potential. [6]. Pervious research predominantly focuses on seedling propagation and cultivation [7,8] bioactive constituents in rhizomes [9,10], biophysiological traits [11–13], pharmacological mechanisms [14]. In contrast, studies addressing provenance-based variation in growth traits and medicinal components remain notably scarce.

According to the IUCN Red List of Threatened Species, *P. cyrtonema* is classified as Near Threatened (NT) [15], which confirms the ongoing decline of its wild populations and the risk of over-exploitation. Thus, it is in a critically endangered state and is in urgent need of protection. Artificial introduction and cultivation are effective ways to protect and preserve germplasm resources of endangered plants [16]. Introduction and cultivation require an understanding of the environmental conditions that affect its growth and the selection of suitable habitat areas to be successful [17]. For medicinal plants, the initial and fundamental step is to screen provenances with high contents of bioactive compounds for domestication and cultivation through provenance trials. This approach is essential to establish a productive and economically viable cultivation model, thereby facilitating scaled-up production. Provenance trials serve as a fundamental methodology in genetic breeding [18]. By evaluating the performance of germplasms from diverse geographical origins under uniform conditions, these trials elucidate patterns of geographic variation, thereby providing a scientific basis for selecting superior provenances and delineating seed zones [19]. This approach has been successfully applied to medicinal plants such as *Astragalus membranaceus* [20], *Dendrobium officinale* [21], and *Panax ginseng* [22], demonstrating significant

efficacy in their genetic improvement. The growing market demand for *P. cyrtonema* has led to overharvesting of wild resources [23], resulting in their alarming depletion and genetic degradation [24].

Given the significant variations in tuber yield and active compound content among *P. cyrtonema* from different production regions and genetic sources [25], coupled with the current scarcity of elite cultivars on the market, conducting provenance trials has become an urgent priority. Although preliminary studies in recent years have screened and classified floral phenotypes [26] and mineral elements [27,28] among different *P. cyrtonema* provenances, most of this work has been largely confined to single-environment observations and analysis of limited medicinal components [29]. A significant research gap remains in the multi-environment evaluation of provenances, particularly regarding the coordinated analysis of medicinal components and growth traits under genotype-by-environment (G × E) interactions.

Therefore, this study utilized 11 geographic provenances of *P. cyrtonema* collected from major production areas across 11 cities in 4 provinces in China. Multi-site cultivation trials were established at three heterogeneous garden trial sites in Zhejiang Province—Longyou (low mountainous basin), Lin'an (mid-high mountainous area), and Jinyun (typical mountainous region). The study objectives are to: (1) quantify the extent of variation in these traits among provenances; (2) identify the key environmental factors driving the observed variation; and (3) select superior provenances suitable for high-yield and high-quality cultivation in Eastern China. This research will provide fundamental data for germplasm conservation and breeding. More importantly, by elucidating the patterns of genotype-by-environment (G × E) interaction, it will offer a scientific basis for developing ecology-based, precise provenances sourcing strategies, thereby directly supporting the efficient and sustainable development of the *P. cyrtonema* industry in Eastern China.

## 2. Materials and methods

### 2.1 Study sites

As shown in Fig 1, the three heterogeneous garden trial sites were located in Longyou County (Quzhou City) (ZJLY), Jinyun County (Lishui City) (ZJJY), and Lin'an District (Hangzhou City) (ZJLA)of Zhejiang Province. The geographic locations and climatic conditions of each site are presented in Table 1. All meteorological data are sourced from the latest official statistical yearbooks compiled by local authorities and meteorological authorities. The field studies did not require specific permits for the experimental sites, as all three locations (ZJLY, ZJJY, and ZJLA) are long-term research bases owned and managed by the Zhejiang Academy of Forestry for scientific purposes.

### 2.2 Plant materials and experimental design

From September to October 2022, wild *P. cyrtonema* rhizomes were collected from 11 provenances across four provinces (Anhui, Zhejiang, Fujian, and Jiangxi) in China (Fig 1). Ninety individual plants were sampled from each provenance, resulting in a total of 990 rhizome samples. Geographical coordinates and altitude for each collection site were recorded in detail (Table 2). All plant materials were authenticated as *P. cyrtonema* by Professor Shi-Ming Cheng of the Zhejiang Academy of Forestry. Each collected rhizome, containing two nodes and one bud (equivalent to a two-year-old growth unit), served as a planting unit. These units were temporarily heeled-in in a greenhouse nursery at the Zhejiang Academy of Forestry prior to the formal experiment.

The field trial was established in late November 2022 at three heterogeneous garden trial sites under bamboo forest canopy with a uniform closure of 0.7. A randomized complete block design was employed with three replications. Each block contained ten plants per provenance. Rhizomes were sown in holes at a planting density of 30 cm × 40 cm (row spacing × plant spacing) and a depth of 8–10 cm. All plants received uniform cultivation management to ensure normal growth.

### 2.3 Experimental procedures

The growth traits of all plants from different provenances at each experimental site were investigated and measured in late June 2023: (1) Plant height was measured to 0.1 cm precision using a tape measure. (2) Stem diameter at 1 cm above

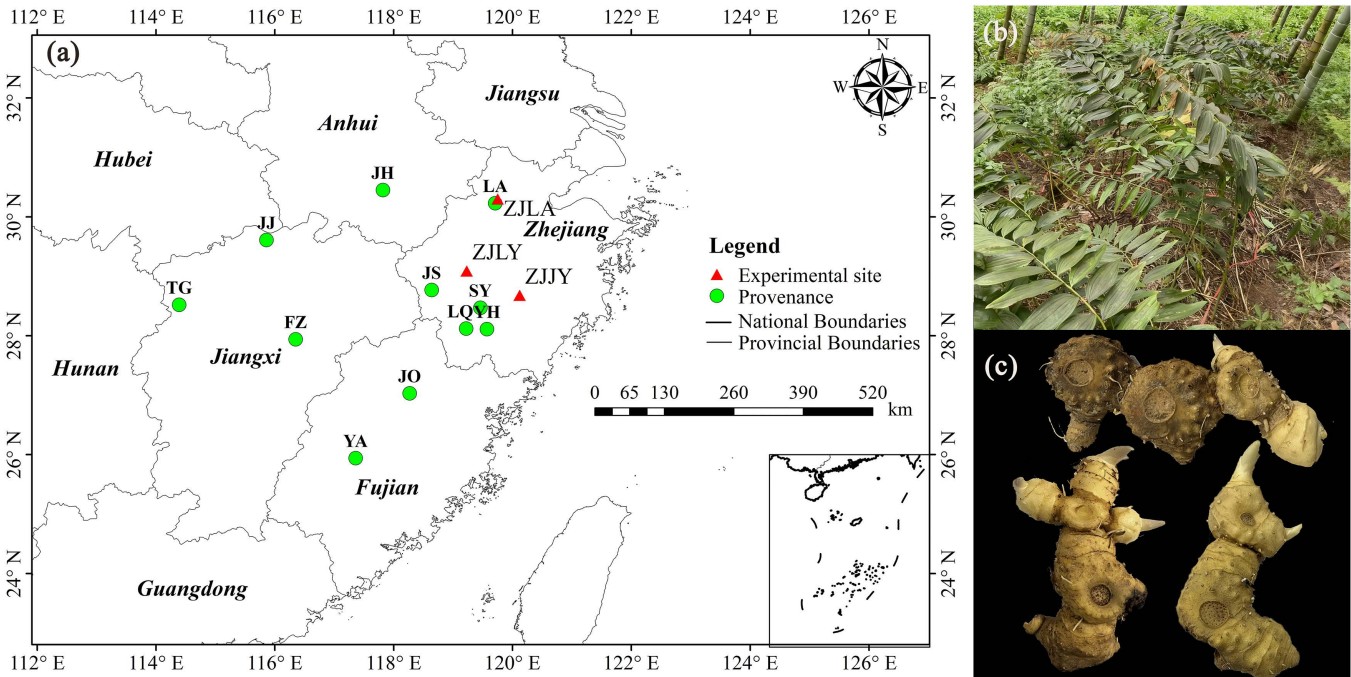

**Fig 1. Distribution map of provenance collection sites and experimental sites for *P. cyrtonema*. (a)** collection sites and experimental sites; **(b)** *P. cyrtonema* in Jinyun sites; **(c)** Rhizomes; The map was created by the authors using geographic coordinates and is for illustrative purposes only.

**Table 1. Geographical location and climatic factors of experimental field.**

| Experimental field | Lon(E)/° | Lat (N)/° | Alt/m | RF/mm | AnnSD/h | AMT/°C |
|---|---|---|---|---|---|---|
| ZJLY | 119.23 | 29.09 | 528 | 1432.9 | 1600.5 | 18.8 |
| ZJJY | 120.12 | 28.68 | 504 | 1156 | 1663.5 | 18.8 |
| ZJLA | 119.75 | 30.30 | 458 | 1318.5 | 1858.2 | 17.5 |

**Note:** Lon, Longitude; Lat, Latitude; Altitude, Alt; RF, Annual rainfall; AnnSD, Annual sunshine duration; AMT, Annual Mean Temperature; The same blow.

**Table 2. Provenance information of *P. cyrtonema*.**

| Code | Provenance (City, Province) | Lon (E)/° | Lat (N)/° | Altitude/m | RF/mm | AnnSD/h | AMT/°C |
|---|---|---|---|---|---|---|---|
| JH | Jiuhua,Anhui | 117.82 | 30.45 | 700 | 1721.7 | 1680.7 | 16.9 |
| JJ | Jiujiang, Jiangxi | 115.86 | 29.61 | 686 | 1783.6 | 1891.5 | 16.5 |
| TG | Tonggu, Jiangxi | 114.39 | 28.52 | 663 | 1771.4 | 1460.4 | 16.4 |
| SY | Songyang, Zhejiang | 119.46 | 28.47 | 645 | 1781.5 | 1840.2 | 17.7 |
| JO | Jianou, Fujian | 118.27 | 27.03 | 545 | 1756.4 | 1644.6 | 18.4 |
| LQ | Longquan, Zhejiang | 119.22 | 28.12 | 564 | 1699.4 | 1849.8 | 17.6 |
| JS | Jiangshan, Zhejiang | 118.64 | 28.77 | 651 | 1648.1 | 2063.3 | 17.1 |
| LA | Lin'an, Zhejiang | 119.71 | 30.23 | 547 | 1613.9 | 1770.9 | 16.7 |
| YH | Yunhe, Zhejiang | 119.57 | 28.11 | 534 | 1638.2 | 1627.2 | 17.9 |
| YA | Yong'an, Fujian | 117.36 | 25.94 | 547 | 1455.5 | 1761.8 | 20.4 |
| FZ | Fuzhou, Jiangxi | 116.35 | 27.94 | 642 | 1781.7 | 1700.4 | 17.9 |

ground was determined to 0.01 cm precision with vernier calipers. (3) Three leaves from the middle canopy of each seedling were selected, and their leaf area, perimeter, length, and width were measured directly using an LAI-2200 leaf area meter (LI-COR, USA), with an accuracy of 0.01 cm for linear dimensions (length, width, perimeter) and 0.01 cm² for leaf area.

In November 2023, rhizomes of all provenances were excavated from the three trial sites and transported to the laboratory for processing. The rhizomes were excised, thoroughly cleaned, and then dried in an oven at 70°C until a constant weight was achieved. The dry weight was measured, and the samples were subsequently used for medicinal component analysis. The polysaccharides, saponins, total phenols and flavonoids, in *P. cyrtonema* rhizome were determined using kits (Suzhou Comin Biotechnology Co., Ltd., Suzhou, China) in accordance with the kits' instructions [30].

The total polysaccharide content was determined using the phenol-sulfuric acid method: approximately 0.05 g of dried sample was weighed, extracted with hot water, and subjected to alcohol precipitation. The supernatant was reacted with phenol-concentrated sulfuric acid reagent at 90°C for 20 minutes, and the absorbance was measured at 490 nm. Glucose was used as the standard for quantification. The total phenol content was determined by the Folin-Ciocalteu method: approximately 0.02 g of dried sample was extracted with 60% ethanol and then reacted with tungstic acid reagent at 25°C for 10 minutes. The absorbance was measured at 760 nm, and gallic acid was used as the standard for calculation. The flavonoid content was measured using the aluminum nitrate colorimetric method: the sample extract was reacted with alkaline aluminum nitrate reagent at 25°C for 15 minutes, and the absorbance was measured at 510 nm. Rutin was used as the standard for calculation. The total saponin content was determined by the vanillin-perchloric acid method: after ultrasonic extraction and evaporation to dryness, the sample was reacted with the coloring reagent at 55°C for 20 minutes, and the absorbance was measured at 589 nm. Oleanolic acid was used as the standard for calculation. All contents were expressed per gram of dry weight (mg·g$^{-1}$).

## 2.4 Data processing

Data entry and organization were performed using Microsoft Excel 2016. All subsequent statistical analyses, including analysis of variance (ANOVA), significance tests, Pearson correlation analysis, principal component analysis (PCA), and cluster analysis, were conducted using SPSS 25.0 (IBM Corp., Armonk, NY, USA).

Prior to PCA and cluster analysis, the raw data were standardized using the formula $Z_{ij}=(X_{ij}-\mu_i)/\sigma_i$, where $Z_{ij}$ is the standardized value of the "i"-th trait for the "j"-th provenance, $X_{ij}$ is the original measured value, $\mu_i$ is the mean of the "i"-th trait, and $\sigma_i$ is the standard deviation of the "i"-th trait.

Analysis of variance (ANOVA) was performed to assess the variation among provenances using the following linear model:

$$y_{ijk} = \mu + P_i + B_j + PB_{ij} + \varepsilon_{ijk}$$

where $y_{ijk}$ was the performance of the kth tree of *i*th provenance growing at *j*th biock, $\mu$ was the overall mean, Pi was the effect of *i*th provenance, $B_j$ was the effect of jth block, $PB_{i(j)}$ was the interactive effect of *i*th provenance and *j*th block, and $\varepsilon_{ijk}$ was the random error.

The phenotypic and genotypic coefficient of variation (PCV and GCV) was calculated using the following formula [31]:

$$PCV = \frac{\sqrt{\sigma_P^2}}{\overline{X}} \times 100$$

$$GCV = \frac{\sqrt{\sigma_g^2}}{\overline{X}} \times 100$$

where $\sigma_P^2$ was the phenotype variance component of the trait, $\sigma_g^2$ was the genetic variance component of the trait and $\overline{X}$ was the average value of the traits.

Provenance repeatability ($h_P^2$) within each site was estimated following Razafimahatratra et al. [32] as follows:

$$h_P^2 = \frac{\sigma_P^2}{\sigma_P^2 + \frac{\sigma_{PB}^2}{B} + \frac{\sigma_e^2}{NB}}$$

where $\sigma_P^2$ was the variance component of the provenance, $\sigma_{PB}^2$ was the variance component of the interaction between the provenance and the block, $\sigma_e^2$ was the variance component of error, B was the number of blocks, N was the number of individuals per provenance in each block.

## 3. Result

### 3.1 Analysis of growth trait variation

A combined analysis of variance (Table 3) of growth traits and medicinal components across different provenances of *P. cyrtonema* at multiple trial sites revealed highly significant differences (P<0.01) among provenances, sites, and in the inter-action between provenances and sites for all traits examined. These results indicate that the same provenance exhibited significant variation in growth traits and medicinal content under different site conditions, while different provenances showed distinct growth patterns even at the same location, demonstrating notable genotype-by-environment interaction effects.

Analysis of growth performance across three heterogeneous garden trial sites revealed distinct patterns among *P. cyrtonema* provenances (Table 4). The ZJJY site recorded the highest mean plant height (87.29 cm), while ZJLY showed the greatest stem diameter (4.33 cm). In contrast, ZJLA exhibited inferior performance, with plant height (58.14 cm) and stem diameter (3.78 cm) measuring only 66.6% and 87.47% of the respective maximum values. Leaf traits also demonstrated considerable variation: the leaf shape index ranged from 3.27 (ZJLY) to 3.98 (ZJLA), leaf area from 26.66 (ZJJY) to 34.83 cm² (ZJLA), and leaf perimeter from 53.13 (ZJLY) to 65.72 cm (ZJJY). Notably, the YH provenance consistently achieved the tallest plant height across all sites (113.09 cm in ZJLY 105.5 cm in ZJJY, and 73.11 cm in ZJLA), exceeding the short-est provenance (FZ, 49.44 cm) by 128.72% in ZJLY. The SY provenance exhibited superior stem diameter in both ZJLY (6.62 cm) and ZJLA (4.47 cm), measuring 2.06 times thicker than FZ in ZJLY.

Medicinal composition analysis showed ZJJY with the highest mean polysaccharide (12.52 mg·g⁻¹) and flavonoid (2.53 mg·g⁻¹) contents, though the lowest total phenolics (3.90 mg·g⁻¹). ZJLY recorded the maximum saponin content (27.15 mg·g⁻¹) but the minimum polysaccharide (9.44 mg·g⁻¹) and flavonoid (1.83 mg·g⁻¹) levels. ZJLA contained the highest total phenolics (4.49 mg·g⁻¹) yet the lowest saponin (18.41 mg·g⁻¹). SY provenance consistently led in polysaccharide content across sites (17.37–21.57 mg·g⁻¹) and total phenolics (>5.28 mg·g⁻¹), while YH excelled in flavonoid accumulation (2.78–3.48 mg·g⁻¹). Conversely, YA and JO provenances performed poorly.

The phenotypic coefficient of variation (PCV) for growth traits ranged from 14.58% to 46.89%, with genotypic coefficient of variation (GCV) between 8.96% and 30.59% (Table 5). Specifically, plant height PCV and GCV varied between 18.08–35.95% and 13.94–30.59%, respectively, with ZJJY showing the lowest and ZJLY the highest values for both parameters. Similarly, stem diameter PCV (14.58–30.18%) and GCV (12.78–23.12%) followed the same pattern with height across sites. For medici-nal components, PCV ranged from 7.41% to 46.77% and GCV from 5.99% to 44.92%. Polysaccharides exhibited substantially higher PCV and GCV than other components, suggesting greater inter-provenance variation, while saponins showed relatively stable accumulation with lower variation coefficients. Notably, GCV values were consistently lower than PCV across all traits and sites. Provenance repeatability ($h_P^2$) ranged from 0.63 to 0.99, indicating high heritability for most traits.

### 3.2 Correlation analysis

Correlation analysis among growth traits, medicinal components, and geo-climatic factors in *P. cyrtonema* is summarized in Fig 2. Significant correlations were observed not only between growth and medicinal traits but also with geo-climatic variables. Plant height showed highly significant positive correlations with stem diameter, polysaccharide content, and

**Table 3. Joint analysis of variance for growth and medicinal traits of provenance at different sites.**

| Type | Trait | Source | SS | df | MS | F | Sig. |
|---|---|---|---|---|---|---|---|
| Growth | Height | Provenance | 174847.39 | 10 | 17484.74 | 121.67 | 0.000 |
| | | Site | 140007.27 | 2 | 70003.63 | 487.12 | 0.000 |
| | | Provenance × site | 92494.58 | 20 | 4624.73 | 32.18 | 0.000 |
| | Stem Diameter | Provenance | 307.44 | 10 | 30.74 | 68.20 | 0.000 |
| | | Site | 55.99 | 2 | 27.99 | 62.10 | 0.000 |
| | | Provenance × site | 190.03 | 20 | 9.50 | 21.08 | 0.000 |
| | Leaf Length | Provenance | 1359.27 | 10 | 135.93 | 32.78 | 0.000 |
| | | Site | 815.06 | 2 | 407.53 | 98.28 | 0.000 |
| | | Provenance × site | 3022.68 | 20 | 151.13 | 36.45 | 0.000 |
| | Leaf Width | Provenance | 68.82 | 10 | 6.88 | 21.88 | 0.000 |
| | | Site | 88.95 | 2 | 44.48 | 141.38 | 0.000 |
| | | Provenance × site | 90.49 | 20 | 4.52 | 14.38 | 0.000 |
| | Leaf Shape Index | Provenance | 48.89 | 10 | 4.89 | 8.09 | 0.000 |
| | | Site | 100.52 | 2 | 50.26 | 83.11 | 0.000 |
| | | Provenance × site | 285.59 | 20 | 14.28 | 23.61 | 0.000 |
| | Leaf Area | Provenance | 23055.22 | 10 | 2305.52 | 48.27 | 0.000 |
| | | Site | 11562.82 | 2 | 5781.41 | 121.06 | 0.000 |
| | | Provenance × site | 32884.47 | 20 | 1644.22 | 34.43 | 0.000 |
| | Leaf Perimeter | Provenance | 10609.17 | 10 | 1060.92 | 9.88 | 0.000 |
| | | Site | 27111.58 | 2 | 13555.79 | 126.21 | 0.000 |
| | | Provenance × site | 54056.50 | 20 | 2702.83 | 25.16 | 0.000 |
| Medicinal | Polysaccharides | Provenance | 1935.39 | 10 | 193.54 | 339.92 | 0.000 |
| | | Site | 179.32 | 2 | 89.66 | 157.47 | 0.000 |
| | | Provenance × site | 251.04 | 20 | 12.55 | 22.05 | 0.000 |
| | Total Phenolics | Provenance | 57.39 | 10 | 5.74 | 94.45 | 0.000 |
| | | Site | 6.71 | 2 | 3.36 | 55.23 | 0.000 |
| | | Provenance × site | 5.43 | 20 | 0.27 | 4.47 | 0.000 |
| | Saponins | Provenance | 811.55 | 10 | 81.15 | 68.29 | 0.000 |
| | | Site | 1262.44 | 2 | 631.22 | 531.18 | 0.000 |
| | | Provenance × site | 1088.49 | 20 | 54.42 | 45.80 | 0.000 |
| | Flavonoids | Provenance | 19.34 | 10 | 1.93 | 86.80 | 0.000 |
| | | Site | 8.11 | 2 | 4.05 | 181.91 | 0.000 |
| | | Provenance × site | 2.22 | 20 | 0.11 | 4.98 | 0.000 |

flavonoids (P < 0.01), and a significant positive correlation with total phenolics (P < 0.05). Stem diameter was significantly positively correlated with leaf width, leaf area and flavonoids, and highly significantly correlated with polysaccharides and total phenolics. Leaf length and leaf area also showed significant positive correlations with polysaccharides, total phenolics, and flavonoids. Moreover, polysaccharides, total phenolics, and flavonoids exhibited highly significant positive correlations with each other, whereas saponin content showed limited correlation with most growth and medicinal traits.

In terms of geo-climatic influences, plant height was highly significantly negatively correlated with latitude and annual AnnSD, and highly significantly positively correlated with AMT, while being significantly positively correlated with altitude. Leaf length correlated positively with latitude and AnnSD, but negatively with altitude and AMT. Leaf width showed a highly significant negative correlation with longitude and a highly significant positive correlation with RF. Leaf shape index was

**Table 4. Analysis of growth traits among provenances of *P. cyrtonema* across experimental sites.**

| Site | Pro | H(cm) | StD(cm) | LL(cm) | LW(cm) | LSI | LA(cm²) | LP(cm) | Po(mg·g⁻¹) | TP(mg·g⁻¹) | Sa(mg·g⁻¹) | Fla(mg·g⁻¹) |
|---|---|---|---|---|---|---|---|---|---|---|---|---|
| ZJLY | JH | 50.05±9.77 | 3.55±0.65 | 12.56±1.41 | 3.95±0.66 | 3.27±0.69 | 32.65±6.4 | 59.77±11.7 | 4.78±0.12 | 3.31±0.4 | 26.88±1.12 | 1.78±0.18 |
| | JJ | 58.78±12.57 | 3.69±0.77 | 12.28±1.7 | 4.14±0.46 | 2.98±0.38 | 35.46±7.54 | 67.1±8.53 | 4.42±0.21 | 3.36±0.15 | 25.33±1.6 | 1.56±0.15 |
| | TG | 80.69±14.43 | 3.81±0.73 | 10.18±1.29 | 3.63±0.47 | 2.83±0.4 | 25.84±5.41 | 44.93±10.72 | 9.4±0.58 | 4.17±0.09 | 28.85±1.37 | 1.81±0.28 |
| | SY | 104.18±13.69 | 6.62±0.96 | 11.94±1.79 | 3.85±0.57 | 3.14±0.59 | 31.78±5.55 | 43.66±8.92 | 17.37±0.43 | 5.34±0.11 | 26.68±0.88 | 1.96±0.39 |
| | JO | 60.13±11.93 | 3.76±0.59 | 11.88±1.84 | 3.2±0.56 | 3.8±0.81 | 26.33±4.79 | 49.58±14.61 | 5.28±0.41 | 2.89±0.35 | 27.37±2 | 1.07±0.12 |
| | LQ | 75.71±13.79 | 4.72±0.93 | 14.24±1.59 | 3.89±0.51 | 3.72±0.64 | 37.5±6.68 | 49.39±10.48 | 12.72±0.57 | 4.35±0.13 | 27.18±1.98 | 2.31±0.21 |
| | JS | 98.66±16.83 | 4.07±0.82 | 13.79±1.74 | 4.54±0.57 | 3.06±0.34 | 43.33±9.9 | 69.11±10.45 | 11.38±0.06 | 4.28±0.21 | 31.56±0.96 | 1.82±0.12 |
| | LA | 71.39±16.15 | 4.83±1.02 | 11.63±1.33 | 3.95±0.71 | 3.04±0.7 | 34.08±6.84 | 52.87±13.37 | 11.15±0.38 | 4.55±0.19 | 27.5±0.82 | 1.9±0.02 |
| | YH | 113.09±17.33 | 5.54±1.24 | 11.88±1.88 | 3.87±0.82 | 3.18±0.78 | 32.73±6.24 | 46.57±10.64 | 14.91±1.57 | 4.9±0.1 | 24.78±1.72 | 2.78±0.07 |
| | YA | 50.82±10.63 | 3.78±0.65 | 12.27±1.38 | 3.51±0.43 | 3.56±0.68 | 29.6±4.16 | 49.13±10.8 | 2.76±0.4 | 2.7±0.12 | 24.93±0.56 | 1.43±0.18 |
| | FZ | 49.44±11.25 | 3.22±0.61 | 11.34±1.8 | 3.4±0.5 | 3.39±0.64 | 26.53±6.17 | 52.35±9.52 | 9.65±0.94 | 4±0.24 | 27.61±0.22 | 1.7±0.2 |
| | Mean | 73.90 | 4.33 | 12.18 | 3.81 | 3.27 | 32.35 | 53.13 | 9.44 | 3.99 | 27.15 | 1.83 |
| ZJJY | JH | 104.21±12.39 | 5.1±0.34 | 11.22±2.52 | 3.08±0.7 | 3.75±0.95 | 31.32±9.48 | 68.07±8.39 | 11.07±0.12 | 4.04±0.17 | 22.42±0.27 | 2.22±0.05 |
| | JJ | 79.94±9.91 | 3.9±0.3 | 8.42±0.99 | 2.79±0.41 | 3.06±0.45 | 16.95±3.67 | 54.15±6.03 | 5.25±0.21 | 3.06±0.35 | 24.91±0.53 | 2.01±0.16 |
| | TG | 74.27±6.97 | 3.78±0.29 | 10.65±1.87 | 2.95±0.44 | 3.74±1.06 | 22.35±4.18 | 66.57±5.66 | 9.57±0.58 | 3.73±0.08 | 24.89±1.44 | 2.47±0.1 |
| | SY | 90.16±14.93 | 5.07±0.33 | 16.64±1.9 | 4.18±0.55 | 4.01±0.44 | 48.88±11.36 | 79.74±4.43 | 21.57±0.43 | 5.28±0.19 | 21.34±0.73 | 3.52±0.06 |
| | JO | 78.94±9.29 | 3.95±0.21 | 12.1±1.99 | 2.73±0.33 | 4.47±0.78 | 23.89±5.65 | 69.99±6.85 | 8.82±0.41 | 2.83±0.1 | 23.64±1.41 | 1.69±0.15 |
| | LQ | 90.42±7.04 | 4.05±0.23 | 11.6±2.11 | 2.77±0.49 | 4.31±1.04 | 22.85±5.87 | 65.39±7.75 | 15.74±0.57 | 4.85±0.06 | 22.39±0.95 | 2.76±0.31 |
| | JS | 104.25±12.33 | 4.12±0.5 | 9±1.52 | 2.94±0.57 | 3.14±0.66 | 19.35±5.56 | 58.03±7.93 | 16.58±0.06 | 4.21±0.01 | 23.29±0.4 | 2.61±0.06 |
| | LA | 77.76±5.52 | 3.9±0.25 | 14.67±2.12 | 2.77±0.43 | 5.45±1.28 | 21.93±6.62 | 65.65±7.87 | 14.59±0.38 | 4.43±0.02 | 24.18±1.74 | 2.87±0.05 |
| | YH | 105.5±14.77 | 5.05±0.28 | 15.69±1.54 | 3.85±0.77 | 4.17±0.62 | 42.15±10.02 | 75.12±5.94 | 19.4±1.57 | 5.05±0.08 | 21.29±1.32 | 3.48±0.13 |
| | YA | 76.35±4.13 | 3.81±0.18 | 8.89±1.38 | 2.66±0.53 | 3.43±0.68 | 17.46±5.09 | 54.39±7.89 | 4.28±0.4 | 2.76±0.06 | 21.05±1.09 | 2±0.08 |
| | FZ | 78.43±3.92 | 3.92±0.16 | 11.08±1.65 | 3.31±0.58 | 3.44±0.79 | 26.19±5.84 | 65.78±5.65 | 10.81±0.94 | 2.67±0.3 | 23.18±0.89 | 2.2±0.08 |
| | Mean | 87.29 | 4.24 | 11.82 | 3.09 | 3.91 | 26.66 | 65.72 | 12.52 | 3.9 | 22.96 | 2.53 |
| ZJLA | JH | 33.81±6.46 | 2.46±0.48 | 11.25±1.64 | 2.99±0.31 | 3.81±0.71 | 18.45±2.97 | 38.19±6.89 | 3.74±0.57 | 4.36±0.2 | 27.05±1.25 | 1.82±0.08 |
| | JJ | 61.96±12.02 | 3.61±0.75 | 14.91±2.78 | 3.32±0.51 | 4.55±0.89 | 35.4±6.89 | 53.36±12.42 | 11.07±0.25 | 3.86±0.43 | 28.16±0.23 | 2.05±0.04 |
| | TG | 61.65±14.17 | 3.62±0.66 | 16.47±1.91 | 3.52±0.46 | 4.74±0.71 | 40.41±7.77 | 68.81±12.22 | 8.39±0.19 | 4.6±0.05 | 27.8±0.97 | 2.22±0.06 |
| | SY | 70.64±14.82 | 4.47±0.71 | 17.43±2.04 | 3.83±0.89 | 4.73±0.93 | 45.91±9.37 | 63.74±12.11 | 16.77±0.4 | 5.7±0.11 | 12.72±0.8 | 2.71±0.06 |
| | JO | 41.35±9.53 | 4.1±0.87 | 12.72±3.29 | 3.65±0.56 | 3.57±1.11 | 32.59±7.04 | 64.59±11.1 | 3.34±0.38 | 3.65±0.24 | 17.56±0.3 | 1.44±0.13 |
| | LQ | 59.91±12.33 | 3.83±0.78 | 13.6±3.11 | 3.31±0.5 | 4.23±1.24 | 31.07±6.06 | 51.82±10.58 | 11.73±0.5 | 5.18±0.65 | 7.87±0.89 | 2.36±0.11 |
| | JS | 69.74±14.42 | 3.84±0.67 | 14.21±2.85 | 3.87±0.42 | 3.68±0.7 | 38.04±7.66 | 58.56±13.81 | 11.53±0.06 | 4.84±0.27 | 24.5±1.46 | 2.37±0.06 |
| | LA | 77.6±12.69 | 4.43±0.87 | 9.02±1.11 | 4.37±0.67 | 2.11±0.39 | 35.37±6.62 | 55.4±14.46 | 12.84±0.53 | 4.56±0.5 | 16.35±0.52 | 2.18±0.09 |
| | YH | 73.11±13.79 | 4.3±0.88 | 15.86±2.84 | 3.8±0.54 | 4.22±0.77 | 41.44±8.97 | 68.25±14.73 | 16.01±1.28 | 5.17±0.13 | 17.87±0.07 | 2.88±0.19 |
| | YA | 49.31±10.58 | 3.35±0.72 | 13.11±3.09 | 3.17±0.49 | 4.16±0.86 | 28.79±5.26 | 53.63±12.99 | 6.52±0.26 | 3.66±0.29 | 15.3±0.44 | 1.79±0.09 |
| | FZ | 41.08±8.05 | 3.63±0.77 | 14.26±2.31 | 3.66±0.63 | 3.98±0.77 | 35.64±7.64 | 54.04±13.06 | 7.55±0.5 | 3.82±0.22 | 7.29±0.69 | 2.09±0.06 |
| | Mean | 58.2 | 3.78 | 13.9 | 3.59 | 3.98 | 34.83 | 57.31 | 9.96 | 4.49 | 18.41 | 2.17 |

Note: H, Hight; StD, Stem Diameter; LL, Leaf Length; LW, Leaf Width; LSI, Leaf Shape Index; LA, Leaf Area; LP, Leaf Perimeter; Po, Polysaccharide; TP, Total Phenolics, Sa, Saponins, Fla, Flavonoids.

**Table 5. The PCV, GCV and h²P values of various traits in different experimental sites.**

| Site | Pro | H | StD | LL | LW | LSI | LA | LP | Po | TP | Sa | Fla |
|------|-----|-----|-----|-----|-----|-----|-----|-----|-----|-----|-----|-----|
| ZJLY | PCV(%) | 35.95 | 30.18 | 16.32 | 18.09 | 21.5 | 26.14 | 26.02 | 46.77 | 15.53 | 7.78 | 25.48 |
|      | GCV(%) | 30.59 | 23.12 | 8.96 | 9.4 | 9.22 | 16.22 | 15.48 | 44.92 | 13.68 | 6.69 | 23.37 |
|      | h²P | 0.98 | 0.97 | 0.93 | 0.92 | 0.87 | 0.95 | 0.92 | 0.92 | 0.77 | 0.74 | 0.87 |
| ZJJY | PCV(%) | 18.08 | 14.58 | 27.98 | 23.89 | 27.87 | 46.89 | 15.89 | 45.29 | 16.48 | 7.41 | 22.56 |
|      | GCV(%) | 13.94 | 12.78 | 23.29 | 15.91 | 17.42 | 38.29 | 11.89 | 43.57 | 14.53 | 5.99 | 21.12 |
|      | h²P | 0.98 | 0.99 | 0.99 | 0.96 | 0.95 | 0.98 | 0.97 | 0.94 | 0.79 | 0.63 | 0.9 |
| ZJLA | PCV(%) | 32.46 | 24.69 | 25.34 | 19.12 | 28.54 | 29.56 | 26.47 | 41.24 | 15.03 | 10.12 | 23.02 |
|      | GCV(%) | 24.48 | 14.64 | 17.07 | 10.56 | 18.33 | 20.84 | 15.14 | 39.01 | 13.46 | 9.76 | 21.46 |
|      | h²P | 0.94 | 0.93 | 0.96 | 0.93 | 0.96 | 0.97 | 0.94 | 0.9 | 0.82 | 0.77 | 0.87 |

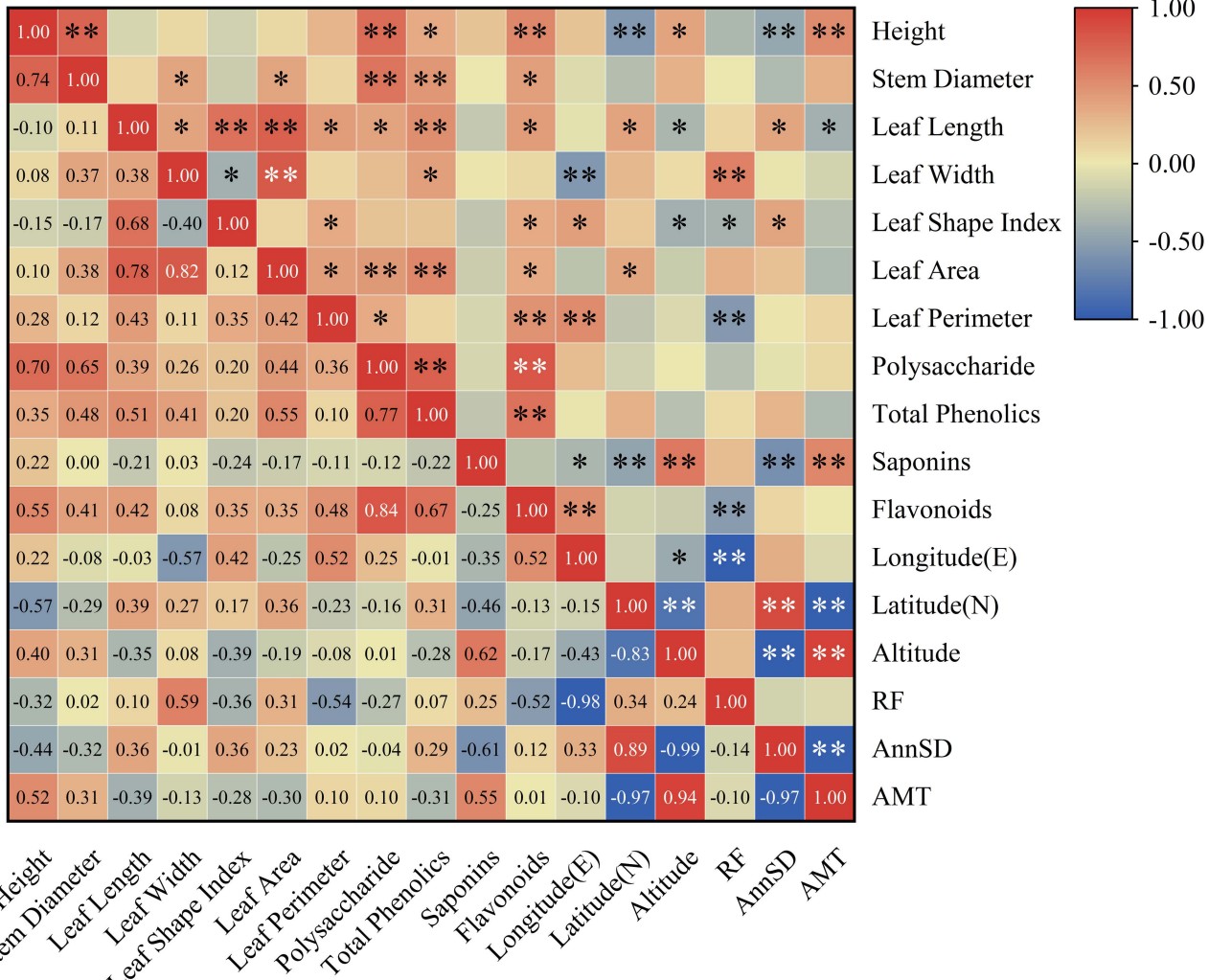

**Fig 2. Correlation analysis between growth, medicinal traits of *P. cyrtonema* and environmental factors. Note:** "*" and "**" indicate significant correlation at the 0.05 level and 0.01 level, respectively.

positively correlated with longitude and AnnSD, and negatively correlated with altitude and RF. Leaf area correlated positively with latitude, whereas leaf perimeter correlated positively with longitude and negatively with RF.

Notably, polysaccharide and total phenolic contents showed no significant correlation with any environmental factor (P > 0.05), suggesting minimal influence of the tested geo-climatic conditions on their accumulation. In contrast, saponin content was highly significantly positively correlated with altitude and AMT, and highly significantly negatively correlated with latitude and AnnSD, with a significant negative correlation with longitude—indicating that warm, high-altitude environments with lower latitude and less sunshine favor saponin accumulation. Flavonoid content was highly significantly positively correlated with longitude and negatively correlated with annual RF, implying that flavonoid synthesis may be promoted in drier regions at higher longitudes.

### 3.6 Principal component analysis (PCA)

PCA of 11 growth and medicinal component indicators across *P. cyrtonema* provenances (Table 6) yielded two principal components (PCs) with variance contribution rates of 68.44% and 16.97%, respectively. The cumulative contribution rate reached 85.41%, exceeded the 80% threshold, indicating these two PCs effectively captured the majority of information from the original 11 variables and were suitable for comprehensive evaluation. PC1 was predominantly loaded by polysaccharides, leaf area, stem diameter, total phenols, leaf length, flavonoids, plant height, leaf width (absolute loading coefficients >0.80), reflecting integrated characteristics of medicinal components (polysaccharide, total phenolics, flavonoids) and growth traits. PC2 was primarily associated with saponins and leaf shape index, representing characteristics of saponins accumulation. Based on the loading coefficients and respective contribution rates of the two PCs, comprehensive scores for each provenance were calculated and ranked (Table 7). The SY and YH provenances achieved the highest rankings, demonstrating superior performance in both growth vigor and medicinal component accumulation. These provenances are recommended as elite genetic resources for *P. cyrtonema* propagation.

### 3.7 Cluster analysis

Hierarchical cluster analysis based on growth traits and medicinal component contents was performed for 11 *P. cyrtonema* provenances using the between-groups linkage method with Euclidean distance., As shown in Fig 3, the provenances were classified into three distinct groups at a Euclidean distance threshold of 15. Cluster I (SY, YH) was characterized by

**Table 6. Principal component analysis based on 11 traits of *P. cyrtonema*.**

| Trait | PC1 | PC2 |
|---|---|---|
| Polysaccharide | 0.98 | −0.01 |
| Leaf Area | 0.97 | 0.43 |
| Stem Diameter | 0.94 | −0.18 |
| Total Phenolics | 0.93 | 0.01 |
| Leaf Length | 0.92 | −0.27 |
| Flavonoids | 0.91 | −0.02 |
| Height | 0.90 | 0.29 |
| Leaf Width | 0.89 | 0.35 |
| Leaf Perimeter | 0.69 | 0.37 |
| Leaf Shape Index | 0.28 | −0.84 |
| Saponins | −0.24 | 0.84 |
| Eigenvalue | 7.53 | 1.86 |
| Contribution rate/% | 68.44 | 16.97 |
| Cumulative contribution rate/% | 68.44 | 85.41 |

**Table 7. Comprehensive evaluation scores and rankings of *P. cyrtonema* provenances.**

| Provenance | PC1 | PC2 | Comprehensive evaluation score | Ranking |
|---|---|---|---|---|
| JH | −2.154 | 0.546 | −1.382 | 9 |
| JJ | −1.831 | 1.224 | −1.046 | 7 |
| TG | −0.688 | 0.853 | −0.326 | 6 |
| SY | 5.137 | −1.01 | 3.345 | 1 |
| JO | −2.199 | −0.707 | −1.625 | 10 |
| LQ | 0.728 | −2.254 | 0.116 | 5 |
| JS | 1.101 | 2.697 | 1.211 | 3 |
| LA | 0.466 | 0.67 | 0.433 | 4 |
| YH | 4.403 | −0.191 | 2.981 | 2 |
| YA | −3.441 | −1.198 | −2.558 | 11 |
| FZ | −1.523 | −0.631 | −1.149 | 8 |

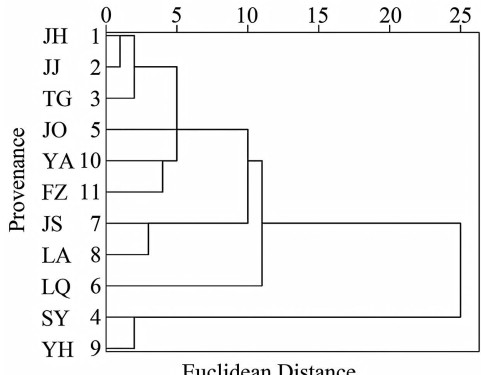

**Fig 3. Dendrogram of cluster analysis for *P. cyrtonema* of different provenances.**

significantly higher contents of major medicinal components and superior growth traits compared to other provenances. Cluster II (JS, LA, LQ) exhibited intermediate levels for both growth traits and medicinal components. Cluster III (JH, JJ, TG, JO, YA, FZ) displayed slender plant morphology, relatively inferior phenotypic performance, and lower contents of major medicinal components. This analysis revealed a discernible geographical regionality in the morphological characteristics among the *P. cyrtonema* provenances.

## 4. Discussion

This study represents the first multi-site provenance trial conducted for *P. cyrtonema*. We established three heterogeneous garden trial sites simultaneously in Longyou, Jinyun, and Lin'an districts of Zhejiang Province, and systematically evaluated growth traits and medicinal components across 11 provenances. Our results demonstrated highly significant differences ($p < 0.01$) among provenances, sites, and in the interaction between provenances and sites for all traits, indicating that these variations are predominantly governed by genetic factors and genotype-by-environment interactions (G×E). These findings align with reports by Jiang et al. [28], Wang et al. [33] and Tao et al. [34], and underscore the importance of selecting superior provenances for cultivation in practical production. Building on this, we further investigated G×E effects and conducted a comprehensive evaluation of the 11 provenances based on both growth traits and medicinal components.

## 4.1 Variations in growth and medical traits

Genetic heritability and variation are fundamental components in plants breeding, serving as crucial parameters for the selection and evaluation of superior provenances [35]. The magnitude of the coefficient of variation reflects the degree of variability within a population [36]. Larger coefficients indicate greater variability, which is more conducive to the selection of superior provenances [37]. In this study, the GCV values for all traits across experimental sites were consistently lower than their corresponding PCV values, aligning with the theoretical expectation that genetic variation constitutes a subset of phenotypic variation. This finding is consistent with previous research reporting that both genetic variance and genetic coefficient of variation for agronomic traits and medicinal components in *P. cyrtonema* are generally lower than their environmental counterparts [33]. Specifically, both phenotypic and genotypic coefficients of variation were relatively large for growth traits (PCV: 14.58%−46.89%) and medicinal components (excluding saponins, PCV: 15.03%−46.77%) across the three trial sites. These results indicate considerable genetic differentiation among provenances in both growth characteristics and medicinal composition, corroborating findings from molecular marker studies by Jiang et al. [38] and Liu et al. [39]. This pronounced genetic differentiation can be attributed to limited gene flow due to geographical isolation in *P. cyrtonema*, combined with its self-pollinating reproductive strategy that naturally restricts cross-pollination and genetic exchange.

Among specific traits, polysaccharides exhibited substantially higher PCV and GCV values compared to other medicinal components, suggesting significant genetic variation in polysaccharide content among provenances and considerable potential for improvement through breeding programs. In contrast, total phenolics demonstrated relatively low PCV and GCV values, indicating stable accumulation across environments. This stability may be attributed to the ecological functions of phenolic compounds as secondary metabolites involved in environmental stress response, defense mechanisms, and signaling. These compounds likely maintain relatively consistent expression patterns within certain ecological parameters as part of the species' adaptive strategies [40].

Provenance repeatability serves as an indicator of trait stability, with higher values suggesting reduced environmental influence and greater phenotypic consistency [41]. Although environmental effects significantly affected trait expression across different sites in our study, the provenance repeatability remained notably high (0.63–0.99). This pattern shows similarities to observations in other medicinal plants, such as *Panax ginseng* (0.63–0.96) [42] and *Picrorhiza kurrooa* (0.65–0.86) [43], indicating substantial heritability for these traits and suggesting significant potential for genetic gain through selection [44]. Growth traits generally demonstrated higher repeatability (mean: 0.95 across all sites) compared to medicinal traits, indicating greater stability in growth characteristics and supporting their utility as primary indicators in geographical variation studies of *P. cyrtonema*. Nevertheless, we acknowledge that these estimates are derived from a single-year assessment. Multi-year longitudinal studies would be valuable to confirm the stability of genetic control across different developmental stages and to account for potential temporal variation in trait expression.

## 4.2 Correlation analysis

This study revealed significant positive correlations between growth indicators (plant height, stem diameter, leaf length, and leaf area) and medicinal components (polysaccharides, flavonoids, and total phenolics) in *P. cyrtonema*, with some relationships reaching highly significant levels. These findings suggest close physiological and ecological linkages between vegetative growth and the accumulation of secondary metabolites in this species. All three heterogeneous garden trial sites in this study were established under bamboo forests, which previous research has identified as having the simplest structure, lowest species composition, and poorest diversity among community types, resulting in reduced interspecific competition pressure [45]. Moreover, the canopy density (0.6) measured in our study aligns with the optimal range for *P. cyrtonema* growth [46]. Consequently, compared to evergreen broad-leaved forests and coniferous-broadleaf mixed forests, bamboo forests provide more suitable growing conditions for *P. cyrtonema*, ultimately yielding superior

quality. Favorable habitats typically support both the growth and development of *P. cyrtonema* and the synthesis of secondary metabolites [47], leading to the synchronized improvement of growth indicators and medicinal component content observed in this study.

These findings further support the concept that the formation of "geo-authenticity" (Daodi) in *P. cyrtonema* essentially represents the collaborative evolution of phenotypic traits and medicinal components under long-term selective pressures from specific geographic and climatic factors. Our study demonstrated that plant height in *P. cyrtonema* positively correlates with mean annual temperature and altitude, while negatively correlating with latitude, consistent with general principles of plant physiology [48]. Warmer, lower-latitude environments typically enable plants to complete their vegetative growth cycle earlier or maintain faster growth rates [49]. The positive correlation between plant height and altitude may be attributed to superior growing conditions at higher elevations within the studied range. Factors such as more intense solar radiation and a larger diurnal temperature span can promote photosynthetic efficiency, thereby facilitating the accumulation of biomass [50]. As the primary site of photosynthesis, leaf phenotypes directly influence photosynthetic area and resource allocation during plant growth. Plants often modify leaf morphological characteristics to better adapt to their habitats, a phenomenon clearly demonstrated in our findings [51]. The significant positive correlations between leaf length and both latitude and annual sunshine hours align with observations in *Xanthoceras sorbifolium* [52], suggesting that *P. cyrtonema* tends to develop.

In contrast to the generally significant correlations observed in growth traits, the contents of polysaccharides and total phenolics in different *P. cyrtonema* provenances showed no significant correlation with the analyzed geographical and environmental factors. This aligns with findings by Su et al. [23], who also reported inconspicuous geographical variation patterns in polysaccharides and extractives of *P. cyrtonema*. This phenomenon may be attributed to strong provenance-by-microenvironment interactions, where the synthesis of polysaccharides and extractives is regulated by more complex, non-linear micro-environmental mechanisms rather than simple linear drives of macro-climatic factors.

In comparison, saponin and flavonoid accumulation in *P. cyrtonema* demonstrated clear environmental drivers. Higher altitude, warmer climate, and lower latitude and longitude were found to favor saponin accumulation, consistent with studies on *Paris polyphylla* [53] and *Bupleurum chinense* [54]. The Jinyun experimental site in this study showed significantly higher saponin content than other sites. Located in the western mountainous area of Zhejiang Province with complex terrain, significant temperature variations, and uneven seasonal precipitation distribution, Jinyun's environmental conditions support the hypothesis that such stress factors stimulate the synthesis of defensive secondary metabolites like saponins [55]. Our results strongly support this theory, indicating that saponins may serve as key chemical defense compounds in *P. cyrtonema*'s adaptation to specific alpine environmental stresses.

The significant negative correlation between flavonoid accumulation and annual rainfall in *P. cyrtonema* resembles patterns observed in *Dendrobium officinale* [56], suggesting that relatively arid conditions may promote flavonoid synthesis. The underlying mechanism involves drought stress typically leading to reactive oxygen species (ROS) accumulation in plants. As important antioxidants, flavonoids are upregulated through activated synthesis pathways to scavenge ROS and protect cellular structures.

In summary, the morphogenesis and medicinal component accumulation in *P. cyrtonema* are driven by environmental factors across different dimensions: growth traits primarily respond to hydrothermal conditions (temperature and rainfall), while key medicinal components are more associated with stress factors (altitude, drought). This finding has significant practical implications for breeding programs. For high saponin content, provenances from high-altitude, warm, low-sunshine regions should be prioritized; for high flavonoid content, provenances from relatively arid regions may be preferable. Notably, both plant height and saponin content showed positive responses to annual temperature, suggesting the potential for synergistic enhancement of growth and saponin accumulation in warm regions, providing valuable screening clues for developing high-yield, high-quality varieties.

## 4.3 Recommended elite provenances

Principal component analysis (PCA) serves as a dimensionality reduction technique to identify the primary factors influencing multiple traits. In this study, PCA was applied to seven growth and four medicinal composition indicators of *P. cyrtonema* provenances, extracting two principal components that effectively minimized interference from overlapping variables among the original traits. These two components collectively accounted for 85.41% of the total variance, capturing the majority of trait-related information in an objective and systematic manner, thereby reducing subjective bias. Using the proportional contribution of each component as a weighting factor for comprehensive evaluation, the top three provenances—SY, YH, and JS—were preliminarily identified, with SY showing the highest potential for development. However, high scores do not necessarily indicate overall superiority. Although SY and YH exhibited significantly higher polysaccharide, total phenolic, and flavonoid contents across the three trial sites, their saponin content was not the highest.

Furthermore, the clustering results of the 11 *P. cyrtonema* provenances demonstrate a certain correlation with their geographical distribution patterns. The Cluster I provenances (SY and YH) are primarily distributed in the mid-elevation hilly areas of southern Zhejiang. Category II provenances (Jiangshan, Lin'an, and Longquan) are situated in areas radiating outward from the Cluster I distribution zone, while Cluster III provenances are geographically more distant from Cluster I. The Cluster I provenances exhibited growth indicators and medicinal component contents consistently above the mean values, demonstrating significant adaptive advantages across all three trial sites, thus qualifying as superior provenances for breeding programs. The clustering of Cluster II with Cluster I at a higher hierarchical level suggests that their trait performance is intermediate between superior and common provenances, making them suitable as secondary genetic resources for crossbreeding with Category I materials to broaden the genetic base.

Notably, Cluster III includes several provenances with exceptionally high saponin content, such as the TG provenance (28.85 mg/g in Longyou) and the JJ provenance (24.91 mg/g in Jinyun and 28.16 mg/g in Lin'an). This trade-off has important practical implications for breeding programs. For breeders targeting general-purpose cultivars with broad-spectrum medicinal quality and robust growth, SY and YH represent the optimal choice. However, for specialized breeding objectives focused specifically on high saponin production—such as developing cultivars for saponin-based pharmaceutical applications—these Cluster III provenances should be prioritized. Furthermore, future cross-breeding strategies could aim to combine the high polysaccharide/flavonoid traits of SY/YH with the high saponin traits of TG or JJ through controlled hybridization, potentially generating elite lines that integrate multiple desirable characteristics. This targeted approach to provenance selection based on specific breeding goals lays a foundation for the diversified utilization of *P. cyrtonema* germplasm resources, consistent with the findings of Peng et al. [57]. Finally, based on our research findings, we have designed a simplified decision matrix for growers in different eco-zones (Table 8).

**Table 8. Simplified decision matrix for growers in different eco-zones.**

| Eco-zone Characteristics | Recommended Provenance(s) | Primary Strengths | Considerations |
| --- | --- | --- | --- |
| High-altitude, dry regions | YH (Yunhe) | Highest flavonoid content, good saponin accumulation, broad adaptation | Moderate polysaccharides |
| High-altitude, high-rainfall regions | SY (Songyang) | Highest polysaccharides and total phenolics, excellent growth vigor | Moderate flavonoids |
| Low-altitude, warm regions | JS (Jiangshan) | High saponin content, good growth performance | Variable across sites |
| Saponin-focused production | TG (Tonggu) or JJ (Jiujiang) | Exceptionally high saponin | Lower in other medicinal components |
| General-purpose cultivation | SY or YH | Balanced medicinal profile, superior growth, stable across environments | Best overall choice for most growers |

## 4.4 Study limitations

In summary, this study systematically evaluated superior provenances of *P. cyrtonema* through multiple analytical approaches, providing preliminary insights into the patterns of provenance variation. This multi-site trial advances prior single-environment work by directly quantifying Genotype-by-Environment (G × E) interactions. This enables a critical distinction between broadly adapted provenances, which ensure reliable yield across diverse sites, and specifically adapted ones optimized for particular conditions or target compounds. Consequently, it provides an ecological-genetic basis for spatially precise cultivation strategies within the heterogeneous understory economy, moving beyond mere cataloging of variation toward predictive and de-risked application.

While our findings establish a critical foundation for genetic improvement and elite breeding programs, certain limitations remain to be addressed in future research: (1) The trial covered only three heterogeneous garden sites in East China, insufficient to capture nationwide genetic diversity across ecoregions, potentially omitting critical regional variations; (2) Complex field factors (soil types, microbiome communities) were not systematically quantified [38], possibly constraining the generalizability of conclusions. (3) Although the current study employed combined ANOVA and PCA to identify superior provenances, future multi-environment trials could adopt stability analysis approaches such as AMMI or GGE biplot to select provenances with broad or specific adaptability. While the one-year harvest in this study serves as an effective early screening strategy for identifying genetically superior provenances, it is important to note that the full economic potential of these selections in understory farming systems can only be realized through multi-year cultivation. In practice, *P. cyrtonema* is typically harvested after 2–3 years to maximize rhizome biomass and medicinal compound accumulation. Therefore, the superior provenances identified here (e.g., Songyang and Yunhe) warrant further validation in long-term field trials to confirm their biomass yield performance at commercial harvest stages. Such extended evaluations will ultimately determine their true economic value for understory cultivation. Future work should expand to representative ecoregions (e.g., southwestern mountains, North China Plain), employing integrated transcriptomic-metabolomic analyses to decipher "gene-environment-metabolite" regulatory networks. To assess their adaptive evolution and establish a sustainable utilization framework, future work should conduct long-term monitoring of different *P. cyrtonema* provenances under climate change scenarios, which must include systematic yield estimation.

## 5. Conclusions

This study firstly systematically investigated the variation patterns in growth traits and medicinal components among 11 provenances of *P. cyrtonema* across three heterogeneous garden sites. The results revealed highly significant differences in all traits among provenances, sites, and their interactions, indicating that these variations are jointly governed by genetic factors and genotype-by-environment interactions. Most traits exhibited substantial coefficients of variation and high provenance repeatability, demonstrating significant potential for selecting superior germplasm. Correlation analysis revealed varying degrees of association among growth traits, medicinal components, and geo-climatic factors. Specifically, higher altitude and warmer climate were conducive to saponin accumulation, while drier conditions promoted flavonoid synthesis. Principal component analysis and cluster analysis collectively identified Songyang and Yunhe as the most promising provenances, exhibiting superior growth and medicinal composition across all trial sites, and thus are recommended as elite materials for breeding programs. This study not only provides a scientific basis for the selective breeding of *P. cyrtonema* but also lays a theoretical foundation for understanding the eco-genetic mechanisms underlying its "geo-authenticity" (Daodi), thereby supporting the sustainable utilization of this valuable medicinal resource in understory farming systems.

## Supporting information

**S1 Table. Growth traits among provenances of *P. cyrtonema* across experimental sites.**
(XLSX)

**S2 Table. Medicinal components among *P. cyrtonema* provenances across experimental sites.**
(XLSX)

## Author contributions

**Conceptualization:** Shiming Cheng.

**Data curation:** Ying Hu, Yan Cheng, Zhiyang Qian, Xiashuo Lei, Xiaodeng Shi.

**Formal analysis:** Ying Hu, Yan Cheng, Xiurong Xu, Xiaodeng Shi.

**Investigation:** Ying Hu, Zhiyang Qian, Xiurong Xu.

**Methodology:** Shiming Cheng, Xiashuo Lei.

**Project administration:** Shiming Cheng, Xiaodeng Shi.

**Supervision:** Shiming Cheng.

**Writing – original draft:** Ying Hu, Zhiyang Qian.

**Writing – review & editing:** Xiaodeng Shi.

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
