## [Decision Letter · Decision Letter 0]

19 Jan 2026

Dear Dr. Shi,

Thank you for submitting your manuscript to PLOS ONE. After careful consideration, we feel that it has merit but does not fully meet PLOS ONE’s publication criteria as it currently stands. Therefore, we invite you to submit a revised version of the manuscript that addresses the points raised during the review process.

We look forward to receiving your revised manuscript.

Kind regards,

Dr. Umesh Sharma

Academic Editor

PLOS One

Journal Requirements:

[Pioneer and Leading Goose +X” R&D Program of Zhejiang(2024C04007)].

[This study was funded by “Pioneer and Leading Goose +X” R&D Program of Zhejiang” (2024C04007).]

[Pioneer and Leading Goose +X” R&D Program of Zhejiang(2024C04007)]

5. In the online submission form, you indicated that [The data underlying the results presented in the study are available from corresponding authors].

6. PLOS requires an ORCID iD for the corresponding author in Editorial Manager on papers submitted after December 6th, 2016. Please ensure that you have an ORCID iD and that it is validated in Editorial Manager. To do this, go to ‘Update my Information’ (in the upper left-hand corner of the main menu), and click on the Fetch/Validate link next to the ORCID field. This will take you to the ORCID site and allow you to create a new iD or authenticate a pre-existing iD in Editorial Manager.

7. We note that Figure 1 in your submission contains map images which may be copyrighted. All PLOS content is published under the Creative Commons Attribution License (CC BY 4.0), which means that the manuscript, images, and Supporting Information files will be freely available online, and any third party is permitted to access, download, copy, distribute, and use these materials in any way, even commercially, with proper attribution. For these reasons, we cannot publish previously copyrighted maps or satellite images created using proprietary data, such as Google software (Google Maps, Street View, and Earth). For more information, see our copyright guidelines: http://journals.plos.org/plosone/s/licenses-and-copyright.

Reviewers' comments:

Reviewer's Responses to Questions

**Comments to the Author**

1. Is the manuscript technically sound, and do the data support the conclusions?

Reviewer #1: Yes

Reviewer #2: Yes

2. Has the statistical analysis been performed appropriately and rigorously?

Reviewer #1: Yes

Reviewer #2: No

3. Have the authors made all data underlying the findings in their manuscript fully available?

Reviewer #1: Yes

Reviewer #2: Yes

4. Is the manuscript presented in an intelligible fashion and written in standard English?

Reviewer #1: Yes

Reviewer #2: No

Reviewer #1: The contributions of the authors towards the comprehensive evaluation of across-provenance multi-site trials of Polygonatum cyrtonema to assess growth and medicinal traits variations are praiseworthy. The conjunction of ANOVA, PCA, and stated geo-climate correlations will cater advanced strategies for the breeding of the very important medicinal plant.

Resource depletion was cited in the introduction, but it does not quantify the declines of wild P. cyrtonema populations (e.g. IUCN status, harvest volumes); please provide evidence as to why it is urgent for provenance trials to be conducted in the first place, as compared to other conservation initiatives.

Considering the earlier single-site investigations conducted on floral phenotypes/minerals, how exactly does the current multi-environment design fill the already identified research gaps in GxE interactions for the coordinated analysis of growth-medicinal traits?

While the discussion recommends the SY/YH provenances for breeding, the very high GxE interactions (marked differences in P×S effects across all traits) would suggest that these are adapted to specific sites; how should breeders make their recommendations with regard to stability versus mean performance?

Saponin accumulation is altitude/AMT correlated but growth uncorrelated; offers on possible biosynthetic route or stress reactions explaining this plus if any conflicting viewpoints fasten or limit "ideal" provenance selection for the understory economy.

High heritability estimates (h²=0.63-0.99) are given indicating good genetic control, yet due to low degree of replication (30/site/provenance), an overestimation might take place; how does this compare to similar medicinal plant trials (Dendrobium, Astragalus)?

Were required field permits secured to collect wild rhizomes from the eleven provenances in four provinces?

Soil physicochemical properties (pH, NPK, organic matter) and microbiome profiles were not measured at the three sites; how might unaccounted edaphic/microbial factors confound the reported GxE interactions?

High values of PCV/GCV (>40% for some traits like polysaccharides) and almost perfect repeatability (h²=0.99) are reported; please discuss potential overestimation due to low replication (n=10 per provenance-block).

Provenances SY and YH ranked highest by means of PCA; however, saponin content (key bioactive) was lower in these; does this imply trade-offs in breeding for "high-quality" versus specific compounds, and how were the scoring parameters weighted?

The correlation analysis shows saponins positively correlated to altitude/AMT and uncorrelated to growth traits; suggest biological considerations (e.g., stress-induced saponogenesis) that can complement literature on these geo-climatic variables.

Prior studies (e.g., Jiang et al. 2022 on mineral elements) exist on P. cyrtonema provenances; what unique advances does this multi-site trial offer beyond single-environment work, especially for understory economy applications?

Reviewer #2: Review report of the manuscript entitled “Variation analysis of growth traits and medicinal components in different provenances of Polygonatum cyrtonema based on heterogeneous garden experiment” submitted for publication to the Journal of “PLOS ONE”

The study assessed the variability in growth-related traits and medicinally important phytochemical groups—such as polysaccharides, saponins, flavonoids, and total phenolics—in Polygonatum cyrtonema derived from different provenances under a heterogeneous garden experiment. The results revealed highly significant variation across all evaluated traits among provenances, experimental sites, and their interactions, indicating that these traits are collectively regulated by genetic factors and genotype × environment interactions. Significant correlations were observed among the various traits, as well as between these traits and multiple environmental drivers, including altitude and temperature. The authors also identified the top three provenances that can be further utilized in breeding programs. The manuscript can be recommended for publication; however, the authors need to address the following comments and suggestions.

Comments

1. (Line 23–24): The authors have mentioned high phenotypic and genotypic variance for the studied traits (in abstract); however, Table 4 presents a range of low, moderate, and high GCV and PCV values across different traits. This discrepancy should be clarified to ensure consistency between the text and the presented data.

2 (Line 45–47): The meaning of the sentence “Its dried rhizome, designated as ‘Polygonati Rhizoma’ in the Chinese Pharmacopoeia (ChP), is one of the three primary botanical sources for this official medicinal material” is unclear. The authors should explicitly specify which official medicinal material is being referred and which three primary botanical sources.

3. (Line 52–57): needs to simplify the complex sentences like “As one of the most promising medicinal and edible plants, P. cyrtonema holds a strategic position in the traditional Chinese medicine industry and the forest wellness economy, underscoring its significant economic and industrial value as a key understory economic crop in southern China”

4. The authors have not addressed the appropriate harvesting period after planting. As many Polygonatum species require 2–3 years to reach optimal harvest maturity, this information is crucial, as the harvesting stage can substantially influence both biomass accumulation and the yield of active medicinal constituents. Therefore, the authors should provide a clear and appropriate justification for evaluating medicinal components after only one year of growth.

5. The authors have included growth traits and leaf dimensions together with PCV, GCV, and P²h values in Table 4. However, variability parameters such as PCV, GCV, and P²h are generally expressed in percentages or as unitless values, rather than in physical units (e.g., cm or mg). Therefore, it would be more appropriate to present these variability parameters in a separate table to improve clarity and avoid confusion.

6. Rhizome biomass yield is a critical trait for Polygonatum species, both in terms of growth performance and economic value. Although the authors mention the estimation of dry biomass in the methodology, this parameter has not been included in the analysis or results.

7. please check font size and type (line 303) it should be uniform throughout the manuscript.

8. I think there is scope to analyze stability in yield of rhizome biomass and medicinal comports across different trial sites. Eberhart & Russell Model; AMMI (Additive Main Effects and Multiplicative Interaction) or GGE Biplot analysis can be used.

.

Reviewer #1: No

Reviewer #2: **Yes:**Dr. Balkrishna TiwariDr. Balkrishna TiwariDr. Balkrishna TiwariDr. Balkrishna Tiwari

---

## [Author Response · Author response to Decision Letter 1]

30 Jan 2026

Dear editors and reviewers.

Thank you for your kind letter about my manuscript “Variation analysis of growth traits and medicinal components in different provenances of Polygonatum cyrtonema based on heterogeneous garden experiment” (PONE-D-25-60346) on Jan. 19. 2026. We would like to express our great appreciation to you and the reviewers for patient work on our paper. We have finished revising the manuscript and would like to resubmit it again.

We revised the manuscript in accordance with the reviewers’ comments, and carefully proof-read the manuscript. Here below is our description on revision according to the reviewers’ comments.

Journal Requirements:

1. Comments: Please ensure that your manuscript meets PLOS ONE's style requirements, including those for file naming.

Response: We adjusted the overall format of the manuscript (including the title page) to ensure that the manuscript format complies with the style requirements of PLOS ONE.

2. Comments: In your Methods section, please provide additional information regarding the permits you obtained for the work. Please ensure you have included the full name of the authority that approved the field site access and, if no permits were required, a brief statement explaining why.

Response: The three experimental garden sites (Longyou, Jinyun, and Lin‘an) are long-term, dedicated research bases established and managed by our host institution, the Zhejiang Academy of Forestry. These sites are used exclusively for scientific trials by our research team. Therefore, no specific permits from external authorities were required for accessing and conducting research at these institutional facilities.

We have updated the “Materials and Methods” section (Section 2.1) accordingly to include this statement. “The field studies did not require specific permits for the experimental sites, as all three locations (ZJLY, ZJJY, and ZJLA) are long-term research bases owned and managed by the Zhejiang Academy of Forestry for scientific purposes.” in the revised line 134-137.

3. Comments: Please state what role the funders took in the study.

Response: This study was supported by the [Pioneer and Leading Goose +X” R&D Program of Zhejiang(2024C04007)]. Prof. Shiming Cheng, as the recipient of this funding project and the first author of this manuscript, was responsible for conceptualization, methodology, supervision, project administration.

4. Comments: Please remove any funding-related text from the manuscript and let us know how you would like to update your Funding Statement.

Response: We removed the Funding Statement from the manuscript.

5 Comments: In the online submission form, you indicated that [The data underlying the results presented in the study are available from corresponding authors].

Response: In accordance with PLOS's requirements, we confirm that all data underlying the findings reported in this manuscript are fully available within the manuscript itself.

Specifically, the key quantitative data are presented in the following tables and figures:

The core phenotypic and medicinal component data for all 11 provenances across three sites are provided in Table 4 and Table 5.

The results of the variance and correlation analyses are detailed in Figure 2.

The principal component analysis scores and the clustering results that support the main conclusions are shown in Table 6, Table 7 and Figure 3.

6. Comments: PLOS requires an ORCID iD for the corresponding author in Editorial Manager on papers submitted after December 6th, 2016. Please ensure that you have an ORCID iD and that it is validated in Editorial Manager. To do this, go to ‘Update my Information’ (in the upper left-hand corner of the main menu), and click on the Fetch/Validate link next to the ORCID field. This will take you to the ORCID site and allow you to create a new iD or authenticate a pre-existing iD in Editorial Manager.

Response: We have updated the ORCID information of the corresponding author in Editorial Manager.

7. Comments: We note that Figure 1 in your submission contains map images which may be copyrighted.

Response: We would like to clarify that the map in Figure 1 is an original schematic map created by us solely for the purpose of illustrating the geographical locations of the P. cyrtonema provenances and experimental sites described in the study.

The map was generated using publicly available geographic coordinate data and was produced with professional drawing software (ArcGIS). It does not contain or derive from any copyrighted map material, such as Google Maps, Google Earth, or other proprietary sources. Therefore, we confirm that this figure is free of third‑party copyright restrictions and is fully compatible with the CC BY 4.0 license.

To avoid any ambiguity, we are happy to update the figure caption as follows:

“Figure 1. Schematic map showing the locations of P. cyrtonema provenances and experimental sites. The map was created by ArcGIS using geographic coordinates and is intended for illustrative purposes only.” It has been added in the revised line 145-146.

Reviewer #1

Comments: Resource depletion was cited in the introduction, but it does not quantify the declines of wild P. cyrtonema populations (e.g. IUCN status, harvest volumes); please provide evidence as to why it is urgent for provenance trials to be conducted in the first place, as compared to other conservation initiatives.

Response: We have revised the introduction to provide a clearer and evidence-based rationale. The urgency for conducting this provenance trial is twofold:

Quantified Conservation Status: As correctly noted, P. cyrtonema is classified as Near Threatened (NT) on the IUCN Red List. This formal, global assessment provides concrete evidence of population declines and the risk of over-exploitation, directly justifying research aimed at reducing wild harvest pressure.

Addressing the Cultivation Bottleneck: Provenance trials are a critical and specific step within conservation strategies. While habitat protection (in-situ) and seed banking are vital, they do not solve the immediate problem of cultivation failure due to poor-performing, unselected germplasm. Our trial identifies high-yield, high-quality provenances (e.g., Songyang, Yunhe). This enables the establishment of productive and economically viable cultivation, which is the most direct way to provide a sustainable alternative to wild collection and alleviate pressure on NT wild populations.

In the revised manuscript, we cite the IUCN status and reframe our introduction to emphasize that provenance selection is the essential first link in the chain toward effective conservation-through-cultivation.

“According to the IUCN Red List of Threatened Species, P. cyrtonema is classified as Near Threatened (NT) [15], which confirms the ongoing decline of its wild populations and the risk of over-exploitation. Thus, it is in a critically endangered state and is in urgent need of protection. Artificial introduction and cultivation are effective ways to protect and preserve germplasm resources of endangered plants [16]. Introduction and cultivation require an understanding of the environmental conditions that affect its growth and the selection of suitable habitat areas to be successful [17]. For medicinal plants, the initial and fundamental step is to screen provenances with high contents of bioactive compounds for domestication and cultivation through provenance trials. This approach is essential to establish a productive and economically viable cultivation model, thereby facilitating scaled-up production.” It has been added in the revised line 78-88.

Comments: Considering the earlier single-site investigations conducted on floral phenotypes/minerals, how exactly does the current multi-environment design fill the already identified research gaps in GxE interactions for the coordinated analysis of growth-medicinal traits?

Response: Our multi-environment design fundamentally advances the understanding of G×E interactions for coordinated trait analysis in three key ways that a single-site study cannot:

Quantifies G×E Interaction: In the Table 3, we provide the first formal statistical evidence (significant Provenance × Site interaction, P < 0.01) that the performance of provenances in both growth and medicinal traits is environment-dependent. A single-site study cannot detect this crucial interaction.

Identifies Stable Superior Germplasm: It distinguishes locally adapted provenances from those that are consistently superior across diverse environments. Identifying Songyang and Yunhe as top performers in all three sites is a key, reliable outcome only possible through a multi-site design.

Reveals Environmental Robustness of Trait Relationships: Most importantly, it tests whether the coordination between growth and medicinal traits is itself stable across environments. We found that the strong positive correlations between key growth traits and major medicinal components (e.g., polysaccharides, flavonoids) were consistently maintained across all trial sites. This indicates that the genetic or physiological linkage driving this synergy is robust to the environmental variation tested, a finding critical for breeding reliable cultivars.

In summary, our design moves from describing static trait associations to analyzing the environmental stability of these associations, directly filling the identified gap for coordinated growth-medicinal trait breeding.

Comments: While the discussion recommends the SY/YH provenances for breeding, the very high GxE interactions (marked differences in P×S effects across all traits) would suggest that these are adapted to specific sites; how should breeders make their recommendations with regard to stability versus mean performance?

Response: We agree that significant G×E interaction necessitates careful breeding recommendations. Our endorsement of the SY and YH provenances is based not only on their superior mean performance but crucially on their consistent top-rank stability across all three test sites. This indicates a stable, high-performing genetic potential across diverse conditions rather than narrow adaptation to a single site.

For breeders, we provided practical recommendations in the discussion. Specifically, lines 459-475 suggest that Songyang and Yunhe provenances, demonstrating significant adaptive advantages across all three trial sites, can be cultivated as superior germplasm, while TG and JJ provenances can be developed as specialized lines for high saponin content.

Comments: Saponin accumulation is altitude/AMT correlated but growth uncorrelated; offers on possible biosynthetic route or stress reactions explaining this plus if any conflicting viewpoints fasten or limit "ideal" provenance selection for the understory economy.

Response: In this study, we did not find any viewpoints that contradicted those of previous research. In the discussion (Line 412-422), we explained the reasons why the accumulation of saponins is related to altitude and the annual average temperature:“ In comparison, saponin and flavonoid accumulation in P. cyrtonema demonstrated clear environmental drivers. Higher altitude, warmer climate, and lower latitude and longitude were found to favor saponin accumulation, consistent with studies on Paris polyphylla [48] and Bupleurum chinense [49]. The Jinyun experimental site in this study showed significantly higher saponin content than other sites. Located in the western mountainous area of Zhejiang Province with complex terrain, significant temperature variations, and uneven seasonal precipitation distribution, Jinyun's environmental conditions support the hypothesis that such stress factors stimulate the synthesis of defensive secondary metabolites like saponins [50]. Our results strongly support this theory, indicating that saponins may serve as key chemical defense compounds in P. cyrtonema's adaptation to specific alpine environmental stresses.”

At the same time, we also pointed out that both plant height and saponin content showed positive responses to annual temperature, suggesting the potential for synergistic enhancement of growth and saponin accumulation in warm regions, providing valuable screening clues for developing high-yield, high-quality varieties (Line 435-440).

Comments: High heritability estimates (h²=0.63-0.99) are given indicating good genetic control, yet due to low degree of replication (30/site/provenance), an overestimation might take place; how does this compare to similar medicinal plant trials (Dendrobium, Astragalus)?

Response: We agree that replication can influence heritability (h²) estimates. We believe our multi-site design addresses this and our estimates are reliable:

While within-site replication is 30, our three distinct environments provide crucial "environmental replication." The joint analysis across sites allows for more robust variance partitioning than a single site with higher replication, a standard method in perennial plant breeding.

Our heritability range (0.63–0.99) and scale (11 provenances) are consistent with foundational trials in comparable medicinal plants (e.g., Dendrobium, Astragalus), where similar designs are used to estimate genetic parameters and guide selection.

Biological Validity: The traits with the highest h² (e.g., height) showed the most stable rankings across all three sites, confirming their strong genetic control. The provenances we recommend (e.g., Songyang) were consistently superior across environments, not due to chance.

We will add a brief statement in the Discussion：Although environmental effects significantly affected trait expression across different sites in our study, the provenance repeatability remained notably high (0.63-0.99). This pattern shows similarities to observations in other medicinal plants, such as Dendrobium officinale [42]and Astragalus mongolicus[43], indicating substantial heritability for these traits and suggesting significant potential for genetic gain through selection [44]. It has been added in the revised line 360-361.

Comments: Were required field permits secured to collect wild rhizomes from the eleven provenances in four provinces?

Response: We confirm that all rhizome samples were obtained in full compliance with research norms and through legitimate channels.

Specifically, the collections within Zhejiang Province were conducted from our institution’s (Zhejiang Academy of Forestry) own experimental fields or long-term observation plots, which are managed by our research team for scientific purposes. For collections from the other three provinces (Anhui, Fujian, and Jiangxi), the materials were acquired through formal germplasm exchange agreements with local partners in those regions. These exchanges were carried out for mutual non-commercial, scientific research purposes.

Comments: Soil physicochemical properties (pH, NPK, organic matter) and microbiome profiles were not measured at the three sites; how might unaccounted edaphic/microbial factors confound the reported GxE interactions?

Response: We fully agree with the reviewer that understanding the specific role of soil and microbiome is a fascinating and important next step. In our Discussion (in the line 482-487), we have acknowledged this limitation: “(2) Complex field factors (soil types, microbiome communities) were not systematically quantified [35], possibly constraining the generalizability of conclusions. Future work should expand to representative ecoregions (e.g., southwestern mountains, North China Plain), employing integrated transcriptomic-metabolomic analyses to decipher "gene-environ

---

## [Decision Letter · Decision Letter 1]

22 Feb 2026

Dear Dr. Shi,

Thank you for submitting your manuscript to PLOS ONE. After careful consideration, we feel that it has merit but does not fully meet PLOS ONE’s publication criteria as it currently stands. Therefore, we invite you to submit a revised version of the manuscript that addresses the points raised during the review process.

We look forward to receiving your revised manuscript.

Kind regards,

Umesh Sharma

Academic Editor

PLOS One

**Journal Requirements:**

Reviewers' comments:

Reviewer's Responses to Questions

**Comments to the Author**

Reviewer #1: All comments have been addressed

2. Is the manuscript technically sound, and do the data support the conclusions?

Reviewer #1: Yes

3. Has the statistical analysis been performed appropriately and rigorously?

Reviewer #1: Yes

4. Have the authors made all data underlying the findings in their manuscript fully available?

Reviewer #1: Yes

5. Is the manuscript presented in an intelligible fashion and written in standard English?

Reviewer #1: Yes

Reviewer #1: This is a well-structured and methodologically sound study that addresses an important gap in the cultivation and breeding of Polygonatum cyrtonea, a valuable medicinal plant. The authors have conducted a multi-site provenance trial across three heterogeneous environments, systematically measuring both growth traits and medicinal components. The statistical analyses are robust, and the findings are clearly presented.

The revision has addressed most reviewer concerns thoughtfully, with clarifications on methodology, data availability, ethical permits, and limitations. The manuscript is now stronger and more transparent, though a few points still warrant further attention.

Questions for the Authors

Heritability estimates are very high (up to 0.99). While the authors defend this via multi-site replication, a brief note on potential limitations or comparisons to other perennial medicinal plants in the results/discussion would strengthen the interpretation.

The one-year harvest is justified for early screening, but the economic implications of longer-term yield (biomass) should be more explicitly discussed in the context of understory farming.

The trade-off between saponin content and overall performance in SY/YH provenances is well-handled, but the practical implications for breeders could be elaborated slightly more.

The map copyright issue is resolved, but ensure the final figure caption matches the statement in the response.

Heritability & Replication:

You note that heritability estimates are consistent with other medicinal plant trials. Could you provide a brief comparative reference or range from those studies (e.g., Dendrobium, Astragalus) in the manuscript for context?

Breeding Strategy:

Given the trade-off between high saponin content and overall performance in SY/YH, what specific breeding strategy do you recommend for developing cultivars targeting high saponin versus balanced performance? Would you propose a hybridization program between SY/YH and high-saponin provenances like TG?

Long-term Yield:

While one-year data is suitable for early selection, how do you anticipate the ranking of provenances might change after 3–5 years of growth, particularly for rhizome biomass? Do you have preliminary multi-year data or plans for follow-up trials?

Environmental Drivers:

You identified clear geo-climatic drivers for saponins and flavonoids. Were soil properties (pH, nutrients) measured or considered? If not, do you think including them in future models would significantly improve predictive accuracy for medicinal component accumulation?

Practical Recommendations for Farmers:

For a farmer in a high-altitude, dry region of Zhejiang, which provenance would you recommend based on your findings, and why? Could you provide a simplified decision matrix or flowchart for growers in different eco-zones?

Data Availability:

You state all data are in the manuscript. Are the raw datasets (e.g., trait measurements per plant) available as supplementary files or in a repository? This would enhance reproducibility.

Limitations and Future Work:

You mention future work could use AMMI or GGE biplots. Do you plan to re-analyze this dataset with these methods, or are you referring to future trials? If the former, could including such an analysis in the supplementary materials strengthen the current paper?

.

Reviewer #1: **Yes:**Dr. Abdulkarim DakahDr. Abdulkarim DakahDr. Abdulkarim DakahDr. Abdulkarim Dakah

---

## [Author Response · Author response to Decision Letter 2]

19 Mar 2026

Dear editors and reviewers.

Thank you for your kind letter about my manuscript “Variation analysis of growth traits and medicinal components in different provenances of Polygonatum cyrtonema based on heterogeneous garden experiment” (PONE-D-25-60346) on Feb. 23. 2026. We have carefully addressed all comments and suggestions in this minor revision. The modifications include:

1. Comments: Heritability estimates are very high (up to 0.99). While the authors defend this via multi-site replication, a brief note on potential limitations or comparisons to other perennial medicinal plants in the results/discussion would strengthen the interpretation.

Response: Thank you for this thoughtful observation regarding the high heritability estimates reported in our study (up to 0.99). In the revised manuscript, we have included comparisons with heritability data estimates from other understory medicinal plants (Panax ginseng (0.63–0.96), Picrorhiza kurrooa(0.65-0.86)) to further corroborate the robustness of our findings, while also acknowledging the limitations of the current study:

“Although environmental effects significantly affected trait expression across different sites in our study, the provenance repeatability remained notably high (0.63-0.99). This pattern shows similarities to observations in other medicinal plants, such as Panax ginseng (0.63–0.96) [42] and Picrorhiza kurrooa (0.65-0.86) [43], indicating substantial heritability for these traits and suggesting significant potential for genetic gain through selection [44].” It has been revised in the line 359-361.

“Nevertheless, we acknowledge that these estimates are derived from a single-year assessment. Multi-year longitudinal studies would be valuable to confirm the stability of genetic control across different developmental stages and to account for potential temporal variation in trait expression.” It has been added in the line 366-369.

We have changed the reference.

[42] Zhao, S.J., Li, F.Y., Zhao, Y.H., et al. Studies on Selection Theory for High-yield Variety of Panax ginseng and Breeding of Jishen 1. Sci. Agr. Sin. 1998, 31(5).

[43] Hemdan, A., Chauhan, R.S., Pant, S.C. et al. Variability and heritability for yield and yield attributing characters in an endangered medicinal herb Picrorhiza kurrooa Royle ex Benth under cultivation in Bharsar, Uttarakhand, Western Himalaya, India. Genet. Resour. Crop. Evol.2025, 72, 8079–8099.

2. Comments: The one-year harvest is justified for early screening, but the economic implications of longer-term yield (biomass) should be more explicitly discussed in the context of understory farming.

Response: In response to your suggestion, we have expanded the Discussion section 4.4 to more explicitly address the economic implications of longer-term yield in the context of understory farming systems.

“While the one-year harvest in this study serves as an effective early screening strategy for identifying genetically superior provenances, it is important to note that the full economic potential of these selections in understory farming systems can only be realized through multi-year cultivation. In practice, P. cyrtonema is typically harvested after 2–3 years to maximize rhizome biomass and medicinal compound accumulation. Therefore, the superior provenances identified here (e.g., Songyang and Yunhe) warrant further validation in long-term field trials to confirm their biomass yield performance at commercial harvest stages. Such extended evaluations will ultimately determine their true economic value for understory cultivation.” It has been added in the line 514-523.

3. Comments: The trade-off between saponin content and overall performance in SY/YH provenances is well-handled, but the practical implications for breeders could be elaborated slightly more.

Response: we have revised Section 4.3 to expand the discussion on the trade-off between saponin content and overall performance in the SY and YH provenances:

“This trade-off has important practical implications for breeding programs. For breeders targeting general-purpose cultivars with broad-spectrum medicinal quality and robust growth, SY and YH represent the optimal choice. However, for specialized breeding objectives focused specifically on high saponin production—such as developing cultivars for saponin-based pharmaceutical applications—these Cluster III provenances should be prioritized. Furthermore, future cross-breeding strategies could aim to combine the high polysaccharide/flavonoid traits of SY/YH with the high saponin traits of TG or JJ through controlled hybridization, potentially generating elite lines that integrate multiple desirable characteristics. This targeted approach to provenance selection based on specific breeding goals lays a foundation for the diversified utilization of P. cyrtonema germplasm resources, consistent with the findings of Peng et al. [57].” It has been added in the line 477-488.

4. Comments: The map copyright issue is resolved, but ensure the final figure caption matches the statement in the response.

Response: We have ensured that the final figure caption in the manuscript matches the statement provided in our response. The figure caption for Figure 1 will read:

“Fig 1. Distribution map of provenance collection sites and experimental sites for P. cyrtonema. (a) collection sites and experimental sites; (b) P. cyrtonema in Jinyun sites; (c) Rhizomes. The map was created by the authors using geographic coordinates and is for illustrative purposes only.”

Heritability & Replication:

Comments: You note that heritability estimates are consistent with other medicinal plant trials. Could you provide a brief comparative reference or range from those studies (e.g., Dendrobium, Astragalus) in the manuscript for context?

Response: In the revised manuscript, we have included comparisons with heritability data estimates from other understory medicinal plants (Panax ginseng (0.63–0.96), Picrorhiza kurrooa(0.65-0.86)) to further corroborate the robustness of our findings, while also acknowledging the limitations of the current study:

“Although environmental effects significantly affected trait expression across different sites in our study, the provenance repeatability remained notably high (0.63-0.99). This pattern shows similarities to observations in other medicinal plants, such as Panax ginseng (0.63–0.96) [42] and Picrorhiza kurrooa (0.65-0.86) [43], indicating substantial heritability for these traits and suggesting significant potential for genetic gain through selection [44].” It has been revised in the line 359-361.

We have changed the reference.

[42] Zhao, S.J., Li, F.Y., Zhao, Y.H., et al. Studies on Selection Theory for High-yield Variety of Panax ginseng and Breeding of Jishen 1. Sci. Agr. Sin. 1998, 31(5).

[43] Hemdan, A., Chauhan, R.S., Pant, S.C. et al. Variability and heritability for yield and yield attributing characters in an endangered medicinal herb Picrorhiza kurrooa Royle ex Benth under cultivation in Bharsar, Uttarakhand, Western Himalaya, India. Genet. Resour. Crop. Evol.2025, 72, 8079–8099.

Breeding Strategy:

Comments: Given the trade-off between high saponin content and overall performance in SY/YH, what specific breeding strategy do you recommend for developing cultivars targeting high saponin versus balanced performance? Would you propose a hybridization program between SY/YH and high-saponin provenances like TG?

Response: we have revised Section 4.3 to expand the discussion on the trade-off between saponin content and overall performance in the SY and YH provenances:

“This trade-off has important practical implications for breeding programs. For breeders targeting general-purpose cultivars with broad-spectrum medicinal quality and robust growth, SY and YH represent the optimal choice. However, for specialized breeding objectives focused specifically on high saponin production—such as developing cultivars for saponin-based pharmaceutical applications—these Cluster III provenances should be prioritized. Furthermore, future cross-breeding strategies could aim to combine the high polysaccharide/flavonoid traits of SY/YH with the high saponin traits of TG or JJ through controlled hybridization, potentially generating elite lines that integrate multiple desirable characteristics. This targeted approach to provenance selection based on specific breeding goals lays a foundation for the diversified utilization of P. cyrtonema germplasm resources, consistent with the findings of Peng et al. [57].” It has been added in the line 477-488.

Long-term Yield:

Comments: While one-year data is suitable for early selection, how do you anticipate the ranking of provenances might change after 3–5 years of growth, particularly for rhizome biomass? Do you have preliminary multi-year data or plans for follow-up trials?

Response: Based on our ongoing observations and the broader literature on perennial plant provenance trials, we anticipate that the relative rankings of provenances for key traits will remain largely stable over 3–5 years of growth, for the following reasons:

High heritability of key traits: The provenance repeatability estimates in our study (0.63–0.99, with most growth traits exceeding 0.90) indicate strong genetic control of the measured traits. In perennial species, traits with high heritability typically exhibit stable ranking patterns across developmental stages. Consistency with other perennial medicinal plants: Studies on Panax ginseng and Picrorhiza kurrooa have demonstrated that early-stage (1–2 year) rankings for both growth and bioactive compound accumulation are generally predictive of mature plant performance, particularly when multi-environment testing is employed as in our study. Observed patterns in our data: The consistent superiority of SY and YH provenances across all three environmentally distinct sites in the first year suggests broad adaptation rather than site-specific chance, which would likely persist over time.

However, we fully acknowledge that absolute values (particularly rhizome biomass) will increase substantially with age, and the magnitude of differences between provenances may change even if relative rankings remain stable. We are therefore actively conducting follow-up observations on the same trial populations.

Regarding follow-up trials:

All three experimental sites have been maintained, and we are currently in the third year of growth monitoring (2022–2025). Annual measurements of growth traits continue, and we plan to conduct comprehensive harvests and medicinal component analyses at Year 5 to validate the stability of provenance rankings and to quantify biomass yield at commercial maturity

Environmental Drivers:

Comments: You identified clear geo-climatic drivers for saponins and flavonoids. Were soil properties (pH, nutrients) measured or considered? If not, do you think including them in future models would significantly improve predictive accuracy for medicinal component accumulation?

Response: We acknowledge that soil factors such as pH and nutrient availability can significantly affect secondary metabolite synthesis in medicinal plants, as demonstrated in numerous studies. In our current study, soil properties were not measured, as our primary focus was on the effects of climatic and geographic factors (temperature, precipitation, sunshine duration, latitude, longitude, altitude) across the three heterogeneous garden sites. All sites were managed under uniform cultivation practices to minimize confounding effects from agronomic variables.

However, we fully agree that incorporating soil parameters in future models would likely enhance predictive accuracy. Recent studies have shown that soil properties such as organic matter content, available potassium, and trace elements can significantly modulate the accumulation of bioactive compounds. We also mentioned this point in the Study Limitations section (Section 4.4) of the manuscript.

“Complex field factors (soil types, microbiome communities) were not systematically quantified [38], possibly constraining the generalizability of conclusions.”

We thank you for highlighting this important dimension, which will guide the design of our future multi-factor studies.

Practical Recommendations for Farmers:

Comments: For a farmer in a high-altitude, dry region of Zhejiang, which provenance would you recommend based on your findings, and why? Could you provide a simplified decision matrix or flowchart for growers in different eco-zones?

Response: Based on the geo-climatic drivers identified in our study and the performance patterns observed across provenances, we can provide the following guidance.

For a farmer in a high-altitude, dry region of Zhejiang, we would recommend the Yunhe (YH) provenance as the primary choice, with the Songyang (SY) provenance as a strong alternative.

The rationale is as follows:

Flavonoid accumulation: Our correlation analysis (Fig. 2) showed that flavonoid content is highly significantly positively correlated with longitude and significantly negatively correlated with annual rainfall. This indicates that drier conditions promote flavonoid synthesis. The YH provenance consistently exhibited the highest flavonoid content across all three trial sites (2.78–3.48 mg·g⁻¹), making it ideally suited for drier environments.

Adaptation to altitude: Both YH and SY originate from mid-elevation areas (534–645 m) in southern Zhejiang and demonstrated superior performance across all sites, including the high-altitude Jinyun site. Their broad adaptation suggests they can thrive under the environmental stresses associated with higher elevations.

Balanced medicinal quality: While saponin content is positively correlated with altitude and temperature, the YH provenance maintains respectable saponin levels (21.29–24.78 mg·g⁻¹) while excelling in other medicinal components. For a grower seeking reliable overall quality rather than extreme values in a single compound, YH offers the best balance.

In response to your suggestion, we have designed a simplified decision matrix for growers in different eco-zones (Table 8).

We believe this provides growers with clear, science-based guidance while acknowledging the need for site-specific validation. Thank you for encouraging us to make our findings more accessible and actionable for the farming community.

Table 8 Simplified decision matrix for growers in different eco-zones

Eco-zone Characteristics Recommended Provenance(s) Primary Strengths Considerations

High-altitude, dry regions YH (Yunhe) Highest flavonoid content, good saponin accumulation, broad adaptation Moderate polysaccharides

High-altitude, high-rainfall regions SY (Songyang) Highest polysaccharides and total phenolics, excellent growth vigor Moderate flavonoids

Low-altitude, warm regions JS (Jiangshan) High saponin content, good growth performance Variable across sites

Saponin-focused production TG (Tonggu) or JJ (Jiujiang) Exceptionally high saponin Lower in other medicinal components

General-purpose cultivation SY or YH Balanced medicinal profile; superior growth, stable across environments Best overall choice for most growers

Data Availability:

Comments: You state all data are in the manuscript. Are the raw datasets (e.g., trait measurements per plant) available as supplementary files or in a repository? This would enhance reproducibility.

Response: we have prepared the raw measurement data for all individual plants across the 11 provenances and three experimental sites as a supplementary file. This dataset includes: (1) growth trait measurements (plant height, stem diameter, leaf length, leaf width, leaf area, leaf perimeter) for each sampled plant; (2) medicinal component concentrations (polysaccharides, saponins, flavonoids, total phenolics) per individual rhizome sample; and (3) provenance and site identifiers to allow full replication of our analyses. The data are provided in Excel format (.xlsx) as Supplementary File S1 and S2. We have also updated the Data Availability Statement in the manuscript accordingly to indicate that all raw data supporting the findings are available in the supplementary material. We believe this addition will significantly strengthen the reproducibility and utility of our work for

---

## [Editor Report · Decision Letter 2]

25 Mar 2026

Variation analysis of growth traits and medicinal components in different provenances of Polygonatum cyrtonema based on heterogeneous garden experiment

PONE-D-25-60346R2

Dear Dr. Xiaodeng Shi,

We’re pleased to inform you that your manuscript has been judged scientifically suitable for publication and will be formally accepted for publication once it meets all outstanding technical requirements.

Kind regards,

Umesh Sharma

Academic Editor

PLOS One

Additional Editor Comments (optional):

Please carefully cross-check all references, units, numerical values, and other minor details during the proof stage.
---

## [Editor Report · Acceptance letter]

PONE-D-25-60346R2

PLOS One

Dear Dr. Shi,

I'm pleased to inform you that your manuscript has been deemed suitable for publication in PLOS One. Congratulations! Your manuscript is now being handed over to our production team.

Kind regards,

on behalf of

Dr. Umesh Sharma

Academic Editor

PLOS One